# Learning Theory and Algorithms for Forecasting Non-Stationary Time Series

**Vitaly Kuznetsov**
Courant Institute
New York, NY 10011
vitaly@cims.nyu.edu

**Mehryar Mohri**
Courant Institute and Google Research
New York, NY 10011
mohri@cims.nyu.edu

## Abstract

We present data-dependent learning bounds for the general scenario of non-stationary non-mixing stochastic processes. Our learning guarantees are expressed in terms of a data-dependent measure of sequential complexity and a discrepancy measure that can be estimated from data under some mild assumptions. We use our learning bounds to devise new algorithms for non-stationary time series forecasting for which we report some preliminary experimental results.

## 1 Introduction

Time series forecasting plays a crucial role in a number of domains ranging from weather forecasting and earthquake prediction to applications in economics and finance. The classical statistical approaches to time series analysis are based on generative models such as the autoregressive moving average (ARMA) models, or their integrated versions (ARIMA) and several other extensions [Engle, 1982, Bollerslev, 1986, Brockwell and Davis, 1986, Box and Jenkins, 1990, Hamilton, 1994]. Most of these models rely on strong assumptions about the noise terms, often assumed to be i.i.d. random variables sampled from a Gaussian distribution, and the guarantees provided in their support are only asymptotic.

An alternative non-parametric approach to time series analysis consists of extending the standard i.i.d. statistical learning theory framework to that of stochastic processes. In much of this work, the process is assumed to be stationary and suitably mixing [Doukhan, 1994]. Early work along this approach consisted of the VC-dimension bounds for binary classification given by Yu [1994] under the assumption of stationarity and $\beta$-mixing. Under the same assumptions, Meir [2000] presented bounds in terms of covering numbers for regression losses and Mohri and Rostamizadeh [2009] proved general data-dependent Rademacher complexity learning bounds. Vidyasagar [1997] showed that PAC learning algorithms in the i.i.d. setting preserve their PAC learning property in the $\beta$-mixing stationary scenario. A similar result was proven by Shalizi and Kontorovitch [2013] for mixtures of $\beta$-mixing processes and by Berti and Rigo [1997] and Pestov [2010] for exchangeable random variables. Alquier and Wintenberger [2010] and Alquier et al. [2014] also established PAC-Bayesian learning guarantees under weak dependence and stationarity.

A number of algorithm-dependent bounds have also been derived for the stationary mixing setting. Lozano et al. [2006] studied the convergence of regularized boosting. Mohri and Rostamizadeh [2010] gave data-dependent generalization bounds for stable algorithms for $\varphi$-mixing and $\beta$-mixing stationary processes. Steinwart and Christmann [2009] proved fast learning rates for regularized algorithms with $\alpha$-mixing stationary sequences and Modha and Masry [1998] gave guarantees for certain classes of models under the same assumptions.

However, stationarity and mixing are often not valid assumptions. For example, even for Markov chains, which are among the most widely used types of stochastic processes in applications, stationarity does not hold unless the Markov chain is started with an equilibrium distribution. Similarly,

long memory models such as ARFIMA, may not be mixing or mixing may be arbitrarily slow [Baillie, 1996]. In fact, it is possible to construct first order autoregressive processes that are not mixing [Andrews, 1983]. Additionally, the mixing assumption is defined only in terms of the distribution of the underlying stochastic process and ignores the loss function and the hypothesis set used. This suggests that mixing may not be the right property to characterize learning in the setting of stochastic processes.

A number of attempts have been made to relax the assumptions of stationarity and mixing. Adams and Nobel [2010] proved asymptotic guarantees for stationary ergodic sequences. Agarwal and Duchi [2013] gave generalization bounds for asymptotically stationary (mixing) processes in the case of stable on-line learning algorithms. Kuznetsov and Mohri [2014] established learning guarantees for fully non-stationary $\beta$- and $\varphi$-mixing processes.

In this paper, we consider the general case of non-stationary non-mixing processes. We are not aware of any prior work providing generalization bounds in this setting. In fact, our bounds appear to be novel even when the process is stationary (but not mixing). The learning guarantees that we present hold for both bounded and unbounded memory models. Deriving generalization bounds for unbounded memory models even in the stationary mixing case was an open question prior to our work [Meir, 2000]. Our guarantees cover the majority of approaches used in practice, including various autoregressive and state space models.

The key ingredients of our generalization bounds are a data-dependent measure of sequential complexity (*expected sequential covering number* or *sequential Rademacher complexity* [Rakhlin et al., 2010]) and a measure of *discrepancy* between the sample and target distributions. Kuznetsov and Mohri [2014] also give generalization bounds in terms of discrepancy. However, unlike the result of Kuznetsov and Mohri [2014], our analysis does not require any mixing assumptions which are hard to verify in practice. More importantly, under some additional mild assumption, the discrepancy measure that we propose can be estimated from data, which leads to data-dependent learning guarantees for non-stationary non-mixing case.

We devise new algorithms for non-stationary time series forecasting that benefit from our data-dependent guarantees. The parameters of generative models such as ARIMA are typically estimated via the maximum likelihood technique, which often leads to non-convex optimization problems. In contrast, our objective is convex and leads to an optimization problem with a unique global solution that can be found efficiently. Another issue with standard generative models is that they address non-stationarity in the data via a *differencing* transformation which does not always lead to a stationary process. In contrast, we address the problem of non-stationarity in a principled way using our learning guarantees.

The rest of this paper is organized as follows. The formal definition of the time series forecasting learning scenario as well as that of several key concepts is given in Section 2. In Section 3, we introduce and prove our new generalization bounds. In Section 4, we give data-dependent learning bounds based on the empirical discrepancy. These results, combined with a novel analysis of kernel-based hypotheses for time series forecasting (Appendix B), are used to devise new forecasting algorithms in Section 5. In Appendix C, we report the results of preliminary experiments using these algorithms.

## 2 Preliminaries

We consider the following general time series prediction setting where the learner receives a realization $(X_1, Y_1), \ldots, (X_T, Y_T)$ of some stochastic process, with $(X_t, Y_t) \in \mathcal{Z} = \mathcal{X} \times \mathcal{Y}$. The objective of the learner is to select out of a specified family $H$ a hypothesis $h \colon \mathcal{X} \to \mathcal{Y}$ that achieves a small generalization error $\mathbb{E}[L(h(X_{T+1}), Y_{T+1})|Z_1, \ldots, Z_T]$ conditioned on observed data, where $L \colon \mathcal{Y} \times \mathcal{Y} \to [0, \infty)$ is a given loss function. The *path-dependent* generalization error that we consider in this work is a finer measure of the generalization ability than the *averaged* generalization error $\mathbb{E}[L(h(X_{T+1}), Y_{T+1})] = \mathbb{E}[\mathbb{E}[L(h(X_{T+1}), Y_{T+1})|Z_1, \ldots, Z_T]]$ since it only takes into consideration the realized history of the stochastic process and does not average over the set of all possible histories. The results that we present in this paper also apply to the setting where the time parameter $t$ can take non-integer values and prediction lag is an arbitrary number $l \geq 0$. That is, the error is defined by $\mathbb{E}[L(h(X_{T+l}), Y_{T+l})|Z_1, \ldots, Z_T]$ but for notational simplicity we set $l = 1$.

Our setup covers a larger number of scenarios commonly used in practice. The case $\mathcal{X} = \mathcal{Y}^p$ corresponds to a large class of autoregressive models. Taking $\mathcal{X} = \cup_{p=1}^{\infty} \mathcal{Y}^p$ leads to growing memory models which, in particular, include state space models. More generally, $\mathcal{X}$ may contain both the history of the process $\{Y_t\}$ and some additional side information.

To simplify the notation, in the rest of the paper, we will use the shorter notation $f(z) = L(h(x), y)$, for any $z = (x, y) \in \mathcal{Z}$ and introduce the family $\mathcal{F} = \{(x, y) \to L(h(x), y) \colon h \in H\}$ containing such functions $f$. We will assume a bounded loss function, that is $|f| \leq M$ for all $f \in \mathcal{F}$ for some $M \in \mathbb{R}_+$. Finally, we will use the shorthand $\mathbf{Z}_a^b$ to denote a sequence of random variables $Z_a, Z_{a+1}, \dots, Z_b$.

The key quantity of interest in the analysis of generalization is the following supremum of the empirical process defined as follows:

$$\Phi(\mathbf{Z}_1^T) = \sup_{f \in \mathcal{F}} \left( \mathbb{E}[f(Z_{T+1}) | \mathbf{Z}_1^T] - \sum_{t=1}^T q_t f(Z_t) \right), \tag{1}$$

where $q_1, \dots, q_T$ are real numbers, which in the standard learning scenarios are chosen to be uniform. In our general setting, different $Z_t$s may follow different distributions, thus distinct weights could be assigned to the errors made on different sample points depending on their relevance to forecasting the future $Z_{T+1}$. The generalization bounds that we present below are for an arbitrary sequence $\mathbf{q} = (q_1, \dots q_T)$ which, in particular, covers the case of uniform weights. Remarkably, our bounds do not even require the non-negativity of $\mathbf{q}$.

Our generalization bounds are expressed in terms of data-dependent measures of sequential complexity such as expected sequential covering number or sequential Rademacher complexity [Rakhlin et al., 2010]. We give a brief overview of the notion of sequential covering number and refer the reader to the aforementioned reference for further details. We adopt the following definition of a complete binary tree: a $\mathcal{Z}$-valued complete binary tree $\mathbf{z}$ is a sequence $(z_1, \dots, z_T)$ of $T$ mappings $z_t \colon \{\pm 1\}^{t-1} \to \mathcal{Z}, t \in [1, T]$. A path in the tree is $\sigma = (\sigma_1, \dots, \sigma_{T-1})$. To simplify the notation we will write $z_t(\boldsymbol{\sigma})$ instead of $z_t(\sigma_1, \dots, \sigma_{t-1})$, even though $z_t$ depends only on the first $t-1$ elements of $\boldsymbol{\sigma}$. The following definition generalizes the classical notion of covering numbers to sequential setting. A set $V$ of $\mathbb{R}$-valued trees of depth $T$ is a *sequential $\alpha$-cover* (with respect to $\mathbf{q}$-weighted $\ell_p$ norm) of a function class $\mathcal{G}$ on a tree $\mathbf{z}$ of depth $T$ if for all $g \in \mathcal{G}$ and all $\boldsymbol{\sigma} \in \{\pm\}^T$, there is $\mathbf{v} \in V$ such that

$$\left( \sum_{t=1}^T \left| \mathbf{v}_t(\boldsymbol{\sigma}) - g(\mathbf{z}_t(\boldsymbol{\sigma})) \right|^p \right)^{\frac{1}{p}} \leq \|\mathbf{q}\|_q^{-1} \alpha,$$

where $\| \cdot \|_q$ is the dual norm. The *(sequential) covering number* $\mathcal{N}_p(\alpha, \mathcal{G}, \mathbf{z})$ of a function class $\mathcal{G}$ on a given tree $\mathbf{z}$ is defined to be the size of the minimal sequential cover. The *maximal covering number* is then taken to be $\mathcal{N}_p(\alpha, \mathcal{G}) = \sup_{\mathbf{z}} \mathcal{N}_p(\alpha, \mathcal{G}, \mathbf{z})$. One can check that in the case of uniform weights this definition coincides with the standard definition of sequential covering numbers. Note that this is a purely combinatorial notion of complexity which ignores the distribution of the process in the given learning problem.

Data-dependent sequential covering numbers can be defined as follows. Given a stochastic process distributed according to the distribution $\mathbf{p}$ with $\mathbf{p}_t(\cdot | \mathbf{z}_1^{t-1})$ denoting the conditional distribution at time $t$, we sample a $\mathcal{Z} \times \mathcal{Z}$-valued tree of depth $T$ according to the following procedure. Draw two independent samples $Z_1, Z_1'$ from $\mathbf{p}_1$: in the left child of the root draw $Z_2, Z_2'$ according to $\mathbf{p}_2(\cdot | Z_1)$ and in the right child according to $\mathbf{p}_2(\cdot | Z_2')$. More generally, for a node that can be reached by a path $(\sigma_1, \dots, \sigma_t)$, we draw $Z_t, Z_t'$ according to $\mathbf{p}_t(\cdot | S_1(\sigma_1), \dots, S_{t-1}(\sigma_{t-1}))$, where $S_t(1) = Z_t$ and $S_t(-1) = Z_t'$. Let $\mathbf{z}$ denote the tree formed using $Z_t$s and define the *expected covering number* to be $\mathbb{E}_{\mathbf{z} \sim T(\mathbf{p})}[\mathcal{N}_p(\alpha, \mathcal{G}, \mathbf{z})]$, where $T(\mathbf{p})$ denotes the distribution of $\mathbf{z}$.

In a similar manner, one can define other measures of complexity such as sequential Rademacher complexity and the Littlestone dimension [Rakhlin et al., 2015] as well as their data-dependent counterparts [Rakhlin et al., 2011].

The final ingredient needed for expressing our learning guarantees is the notion of *discrepancy* between target distribution and the distribution of the sample:

$$\Delta = \sup_{f \in \mathcal{F}} \left( \mathbb{E}[f(Z_{T+1})|\mathbf{Z}_1^T] - \sum_{t=1}^{T} q_t \, \mathbb{E}[f(Z_t)|\mathbf{Z}_1^{t-1}] \right). \tag{2}$$

The discrepancy $\Delta$ is a natural measure of the non-stationarity of the stochastic process $\mathbf{Z}$ with respect to both the loss function $L$ and the hypothesis set $H$. In particular, note that if the process $\mathbf{Z}$ is i.i.d., then we simply have $\Delta = 0$ provided that $q_t$s form a probability distribution. It is also possible to give bounds on $\Delta$ in terms of other natural distances between distribution. For instance, Pinsker's inequality yields

$$\Delta \le M \left\| \mathbf{P}_{T+1}(\cdot|\mathbf{Z}_1^T) - \sum_{t=1}^{T} q_t \mathbf{P}_t(\cdot|\mathbf{Z}_1^{t-1}) \right\|_{\mathrm{TV}} \le \sqrt{\tfrac{1}{2} D \left( \mathbf{P}_{T+1}(\cdot|\mathbf{Z}_1^T) \,\|\, \sum_{t=1}^{T} q_t \mathbf{P}_t(\cdot|\mathbf{Z}_1^{t-1}) \right)},$$

where $\| \cdot \|_{\mathrm{TV}}$ is the total variation distance and $D(\cdot \,\|\, \cdot)$ the relative entropy, $\mathbf{P}_{t+1}(\cdot|\mathbf{Z}_1^t)$ the conditional distribution of $Z_{t+1}$, and $\sum_{t=1}^{T} q_t \mathbf{P}_t(\cdot|\mathbf{Z}_1^{t-1})$ the mixture of the sample marginals. Alternatively, if the target distribution at lag $l$, $\mathbf{P} = \mathbf{P}_{T+l}$ is a stationary distribution of an asymptotically stationary process $\mathbf{Z}$ [Agarwal and Duchi, 2013, Kuznetsov and Mohri, 2014], then for $q_t = 1/T$ we have

$$\Delta \le \frac{M}{T} \sum_{t=1}^{T} \| \mathbf{P} - \mathbf{P}_{t+l}(\cdot|\mathbf{Z}_{-\infty}^t) \|_{\mathrm{TV}} \le \phi(l),$$

where $\phi(l) = \sup_s \sup_{\mathbf{z}} [\| \mathbf{P} - \mathbf{P}_{l+s}(\cdot|\mathbf{z}_{-\infty}^s) \|_{\mathrm{TV}}]$ is the coefficient of asymptotic stationarity. The process is asymptotically stationary if $\lim_{l \to \infty} \phi(l) = 0$. However, the most important property of the discrepancy $\Delta$ is that, as shown later in Section 4, it can be estimated from data under some additional mild assumptions. [Kuznetsov and Mohri, 2014] also give generalization bounds for non-stationary mixing processes in terms of a related notion of discrepancy. It is not known if the discrepancy measure used in [Kuznetsov and Mohri, 2014] can be estimated from data.

## 3 Generalization Bounds

In this section, we prove new generalization bounds for forecasting non-stationary time series. The first step consists of using *decoupled tangent* sequences to establish concentration results for the supremum of the empirical process $\Phi(\mathbf{Z}_1^T)$. Given a sequence of random variables $\mathbf{Z}_1^T$ we say that $\mathbf{Z'}_1^T$ is a decoupled tangent sequence if $Z_t'$ is distributed according to $\mathbb{P}(\cdot|\mathbf{Z}_1^{t-1})$ and is independent of $\mathbf{Z}_t^\infty$. It is always possible to construct such a sequence of random variables [De la Peña and Giné, 1999]. The next theorem is the main result of this section.

**Theorem 1.** *Let $\mathbf{Z}_1^T$ be a sequence of random variables distributed according to $\mathbf{p}$. Fix $\epsilon > 2\alpha > 0$. Then, the following holds:*

$$\mathbb{P}\big(\Phi(\mathbf{Z}_1^T) - \Delta \ge \epsilon\big) \le \mathop{\mathbb{E}}_{\mathbf{v} \sim T(\mathbf{p})} \big[ \mathcal{N}_1(\alpha, \mathcal{F}, \mathbf{v}) \big] \exp\left( -\frac{(\epsilon - 2\alpha)^2}{2M^2 \|\mathbf{q}\|_2^2} \right).$$

*Proof.* The first step is to observe that, since the difference of the suprema is upper bounded by the supremum of the difference, it suffices to bound the probability of the following event

$$\left\{ \sup_{f \in \mathcal{F}} \left( \sum_{t=1}^{T} q_t (\mathbb{E}[f(Z_t)|\mathbf{Z}_1^{t-1}] - f(Z_t)) \right) \ge \epsilon \right\}.$$

By Markov's inequality, for any $\lambda > 0$, the following inequality holds:

$$\mathbb{P}\left( \sup_{f \in \mathcal{F}} \left( \sum_{t=1}^{T} q_t (\mathbb{E}[f(Z_t)|\mathbf{Z}_1^{t-1}] - f(Z_t)) \right) \ge \epsilon \right)$$

$$\le \exp(-\lambda \epsilon) \, \mathbb{E}\left[ \exp\left( \lambda \sup_{f \in \mathcal{F}} \left( \sum_{t=1}^{T} q_t (\mathbb{E}[f(Z_t)|\mathbf{Z}_1^{t-1}] - f(Z_t)) \right) \right) \right].$$

Since $\mathbf{Z'}_1^T$ is a tangent sequence the following equalities hold: $\mathbb{E}[f(Z_t)|\mathbf{Z}_1^{t-1}] = \mathbb{E}[f(Z'_t)|\mathbf{Z}_1^{t-1}] = \mathbb{E}[f(Z'_t)|\mathbf{Z}_1^T]$. Using these equalities and Jensen's inequality, we obtain the following:

$$\mathbb{E}\left[\exp\left(\lambda\sup_{f\in\mathcal{F}}\sum_{t=1}^T q_t\big(\mathbb{E}[f(Z_t)|\mathbf{Z}_1^{t-1}] - f(Z_t))\big)\right)\right]$$

$$= \mathbb{E}\left[\exp\left(\lambda\sup_{f\in\mathcal{F}}\mathbb{E}\Big[\sum_{t=1}^T q_t\big(f(Z'_t) - f(Z_t))|\mathbf{Z}_1^T\Big]\right)\right]$$

$$\leq \mathbb{E}\left[\exp\left(\lambda\sup_{f\in\mathcal{F}}\sum_{t=1}^T q_t\big(f(Z'_t) - f(Z_t))\big)\right)\right],$$

where the last expectation is taken over the joint measure of $\mathbf{Z}_1^T$ and $\mathbf{Z'}_1^T$. Applying Lemma 5 (Appendix A), we can further bound this expectation by

$$\mathbb{E}_{(\mathbf{z},\mathbf{z'})\sim T(\mathbf{p})}\mathbb{E}_\sigma\left[\exp\left(\lambda\sup_{f\in\mathcal{F}}\sum_{t=1}^T \sigma_t q_t\Big(f(\mathbf{z'}_t(\boldsymbol{\sigma})) - f(\mathbf{z}_t(\boldsymbol{\sigma}))\Big)\right)\right]$$

$$\leq \mathbb{E}_{(\mathbf{z},\mathbf{z'})\sim T(\mathbf{p})}\mathbb{E}_\sigma\left[\exp\left(\lambda\sup_{f\in\mathcal{F}}\sum_{t=1}^T \sigma_t q_t f(\mathbf{z'}_t(\boldsymbol{\sigma})) + \lambda\sup_{f\in\mathcal{F}}\sum_{t=1}^T -\sigma_t q_t f(\mathbf{z}_t(\boldsymbol{\sigma}))\right)\right]$$

$$\leq \tfrac{1}{2}\mathbb{E}_{(\mathbf{z},\mathbf{z'})}\mathbb{E}_\sigma\left[\exp\left(2\lambda\sup_{f\in\mathcal{F}}\sum_{t=1}^T \sigma_t q_t f(\mathbf{z'}_t(\boldsymbol{\sigma}))\right)\right] + \tfrac{1}{2}\mathbb{E}_{(\mathbf{z},\mathbf{z'})}\mathbb{E}_\sigma\left[\exp\left(2\lambda\sup_{f\in\mathcal{F}}\sum_{t=1}^T \sigma_t q_t f(\mathbf{z}_t(\boldsymbol{\sigma}))\right)\right]$$

$$= \mathbb{E}_{\mathbf{z}\sim T(\mathbf{p})}\mathbb{E}_\sigma\left[\exp\left(2\lambda\sup_{f\in\mathcal{F}}\sum_{t=1}^T \sigma_t q_t f(\mathbf{z}_t(\boldsymbol{\sigma}))\right)\right],$$

where for the second inequality we used Young's inequality and for the last equality we used symmetry. Given $\mathbf{z}$ let $C$ denote the minimal $\alpha$-cover with respect to the $\mathbf{q}$-weighted $\ell_1$-norm of $\mathcal{F}$ on $\mathbf{z}$. Then, the following bound holds

$$\sup_{f\in\mathcal{F}}\sum_{t=1}^T \sigma_t q_t f(\mathbf{z}_t(\boldsymbol{\sigma})) \leq \max_{\mathbf{c}\in C}\sum_{t=1}^T \sigma_t q_t \mathbf{c}_t(\boldsymbol{\sigma}) + \alpha.$$

By the monotonicity of the exponential function,

$$\mathbb{E}_\sigma\left[\exp\left(2\lambda\sup_{f\in\mathcal{F}}\sum_{t=1}^T \sigma_t q_t f(\mathbf{z}_t(\boldsymbol{\sigma}))\right)\right] \leq \exp(2\lambda\alpha)\mathbb{E}_\sigma\left[\exp\left(2\lambda\max_{\mathbf{c}\in C}\sum_{t=1}^T \sigma_t q_t \mathbf{c}_t(\boldsymbol{\sigma})\right)\right]$$

$$\leq \exp(2\lambda\alpha)\sum_{\mathbf{c}\in C}\mathbb{E}_\sigma\left[\exp\left(2\lambda\sum_{t=1}^T \sigma_t q_t \mathbf{c}_t(\boldsymbol{\sigma})\right)\right].$$

Since $\mathbf{c}_t(\boldsymbol{\sigma})$ depends only on $\sigma_1,\ldots,\sigma_{T-1}$, by Hoeffding's bound,

$$\mathbb{E}_\sigma\left[\exp\left(2\lambda\sum_{t=1}^T \sigma_t q_t \mathbf{c}_t(\boldsymbol{\sigma})\right)\right] = \mathbb{E}\left[\exp\left(2\lambda\sum_{t=1}^{T-1} \sigma_t q_t \mathbf{c}_t(\boldsymbol{\sigma})\right)\mathbb{E}_{\sigma_T}\left[\exp\left(2\lambda\sigma_T q_T \mathbf{c}_T(\boldsymbol{\sigma})\right)\Big|\boldsymbol{\sigma}_1^{T-1}\right]\right]$$

$$\leq \mathbb{E}\left[\exp\left(2\lambda\sum_{t=1}^{T-1} \sigma_t q_t \mathbf{c}_t(\boldsymbol{\sigma})\right)\exp(2\lambda^2 q_T^2 M^2)\right]$$

and iterating this inequality and using the union bound, we obtain the following:

$$\mathbb{P}\left(\sup_{f\in\mathcal{F}}\sum_{t=1}^T q_t(\mathbb{E}[f(Z_t)|\mathbf{Z}_1^{t-1}] - f(Z_t)) \geq \epsilon\right) \leq \mathbb{E}_{\mathbf{v}\sim T(\mathbf{p})}[\mathcal{N}_1(\alpha,\mathcal{G},\mathbf{v})]\exp\left(-\lambda(\epsilon-2\alpha)+2\lambda^2 M^2\|\mathbf{q}\|_2^2\right).$$

Optimizing over $\lambda$ completes the proof. $\qquad\square$

An immediate consequence of Theorem 1 is the following result.

**Corollary 2.** *For any $\delta > 0$, with probability at least $1 - \delta$, for all $f \in \mathcal{F}$ and all $\alpha > 0$,*

$$\mathbb{E}[f(Z_{T+1})|\mathbf{Z}_1^T] \leq \sum_{t=1}^{T} q_t f(Z_t) + \Delta + 2\alpha + M\|\mathbf{q}\|_2 \sqrt{2 \log \frac{\mathbb{E}_{\mathbf{v} \sim T(\mathbb{P})}[\mathcal{N}_1(\alpha, \mathcal{G}, \mathbf{v})]}{\delta}}.$$

We are not aware of other finite sample bounds in a non-stationary non-mixing case. In fact, our bounds appear to be novel even in the stationary non-mixing case. Using chaining techniques bounds, Theorem 1 and Corollary 2 can be further improved and we will present these results in the full version of this paper.

While Rakhlin et al. [2015] give high probability bounds for a different quantity than the quantity of interest in time series prediction,

$$\sup_{f \in \mathcal{F}} \left( \sum_{t=1}^{T} q_t (\mathbb{E}[f(Z_t)|\mathbf{Z}_1^{t-1}] - f(Z_t)) \right), \tag{3}$$

their analysis of this quantity can also be used in our context to derive high probability bounds for $\Phi(\mathbf{Z}_1^T) - \Delta$. However, this approach results in bounds that are in terms of purely combinatorial notions such as maximal sequential covering numbers $\mathcal{N}_1(\alpha, \mathcal{F})$. While at first sight, this may seem as a minor technical detail, the distinction is crucial in the setting of time series prediction. Consider the following example. Let $Z_1$ be drawn from a uniform distribution on $\{0, 1\}$ and $Z_t \sim p(\cdot|Z_{t-1})$ with $p(\cdot|y)$ being a distribution over $\{0, 1\}$ such that $p(x|y) = 2/3$ if $x = y$ and $1/3$ otherwise. Let $\mathcal{G}$ be defined by $\mathcal{G} = \{g(x) = \mathbf{1}_{x \geq \theta} : \theta \in [0, 1]\}$. Then, one can check that $\mathbb{E}_{\mathbf{v} \sim T(\mathbb{P})}[\mathcal{N}_1(\alpha, \mathcal{G}, \mathbf{v})] = 2$, while $\mathcal{N}_1(\alpha, \mathcal{G}) \geq 2^T$. The data-dependent bounds of Theorem 1 and Corollary 2 highlight the fact that the task of time series prediction lies in between the familiar i.i.d. scenario and adversarial on-line learning setting.

However, the key component of our learning guarantees is the discrepancy term $\Delta$. Note that in the general non-stationary case, the bounds of Theorem 1 may not converge to zero due to the discrepancy between the target and sample distributions. This is also consistent with the lower bounds of Barve and Long [1996] that we discuss in more detail in Section 4. However, convergence can be established in some special cases. In the i.i.d. case our bounds reduce to the standard covering numbers learning guarantees. In the drifting scenario, with $\mathbf{Z}_1^T$ being a sequence of independent random variables, our discrepancy measure coincides with the one used and studied in [Mohri and Muñoz Medina, 2012]. Convergence can also be established in asymptotically stationary and stationary mixing cases. However, as we show in Section 4, the most important advantage of our bounds is that the discrepancy measure we use can be estimated from data.

## 4 Estimating Discrepancy

In Section 3, we showed that the discrepancy $\Delta$ is crucial for forecasting non-stationary time series. In particular, if we could select a distribution $\mathbf{q}$ over the sample $\mathbf{Z}_1^T$ that would minimize the discrepancy $\Delta$ and use it to weight training points, then we would have a better learning guarantee for an algorithm trained on this weighted sample. In some special cases, the discrepancy $\Delta$ can be computed analytically. However, in general, we do not have access to the distribution of $\mathbf{Z}_1^T$ and hence we need to estimate the discrepancy from the data. Furthermore, in practice, we never observe $Z_{T+1}$ and it is not possible to estimate $\Delta$ without some further assumptions. One natural assumption is that the distribution $\mathbf{P}_t$ of $Z_t$ does not change drastically with $t$ on average. Under this assumption the last $s$ observations $\mathbf{Z}_{T-s+1}^T$ are effectively drawn from the distribution close to $\mathbf{P}_{T+1}$. More precisely, we can write

$$\Delta \leq \sup_{f \in \mathcal{F}} \left( \frac{1}{s} \sum_{t=T-s+1}^{T} \mathbb{E}[f(Z_t)|\mathbf{Z}_1^{t-1}] - \sum_{t=1}^{T} q_t \, \mathbb{E}[f(Z_t)|\mathbf{Z}_1^{t-1}] \right)$$

$$+ \sup_{f \in \mathcal{F}} \left( \mathbb{E}[f(Z_{T+1})|\mathbf{Z}_1^T] - \frac{1}{s} \sum_{t=T-s+1}^{T} \mathbb{E}[f(Z_t)|\mathbf{Z}_1^{t-1}] \right).$$

We will assume that the second term, denoted by $\Delta_s$, is sufficiently small and will show that the first term can be estimated from data. But, we first note that our assumption is necessary for learning in

this setting. Observe that

$$\sup_{f \in \mathcal{F}} \Big( \mathbb{E}[Z_{T+1}|\mathbf{Z}_1^T] - \mathbb{E}[f(Z_r)|\mathbf{Z}_1^{r-1}] \Big) \leq \sum_{t=r}^{T} \sup_{f \in \mathcal{F}} \Big( \mathbb{E}[f(Z_{t+1})|\mathbf{Z}_1^t] - \mathbb{E}[f(Z_t)|\mathbf{Z}_1^{t-1}] \Big)$$

$$\leq M \sum_{t=r}^{T} \|\mathbf{P}_{t+1}(\cdot|\mathbf{Z}_1^t) - \mathbf{P}_t(\cdot|\mathbf{Z}_1^{t-1})\|_{\mathrm{TV}},$$

for all $r = T - s + 1, \dots, T$. Therefore, we must have

$$\Delta_s \leq \frac{1}{s} \sum_{t=T-s+1}^{T} \sup_{f \in \mathcal{F}} \Big( \mathbb{E}[Z_{T+1}|\mathbf{Z}_1^T] - \mathbb{E}[f(Z_t)|\mathbf{Z}_1^t] \Big) \leq \frac{s+1}{2} M\gamma,$$

where $\gamma = \sup_t \|\mathbf{P}_{t+1}(\cdot|\mathbf{Z}_1^t) - \mathbf{P}_t(\cdot|\mathbf{Z}_1^{t-1})\|_{\mathrm{TV}}$. Barve and Long [1996] showed that $[\text{VC-dim}(H)\gamma]^{\frac{1}{3}}$ is a lower bound on the generalization error in the setting of binary classification where $\mathbf{Z}_1^T$ is a sequence of independent but not identically distributed random variables (drifting). This setting is a special case of the more general scenario that we are considering.

The following result shows that we can estimate the first term in the upper bound on $\Delta$.

**Theorem 3.** *Let $\mathbf{Z}_1^T$ be a sequence of random variables. Then, for any $\delta > 0$, with probability at least $1 - \delta$, the following holds for all $\alpha > 0$:*

$$\sup_{f \in \mathcal{F}} \left( \sum_{t=1}^{T} (p_t - q_t) \, \mathbb{E}[f(Z_t)|\mathbf{Z}_1^{t-1}] \right) \leq \sup_{f \in \mathcal{F}} \left( \sum_{t=1}^{T} (p_t - q_t) f(Z_t) \right) + B,$$

*where $B = 2\alpha + M\|\mathbf{q} - \mathbf{p}\|_2 \sqrt{2 \log \frac{\mathbb{E}_{\mathbf{z} \sim T(\mathbf{p})}[\mathcal{N}_1(\alpha, \mathcal{G}, \mathbf{z})]}{\delta}}$ and where $\mathbf{p}$ is the uniform distribution over the last $s$ points.*

The proof of this result is given in Appendix A. Theorem 1 and Theorem 3 combined with the union bound yield the following result.

**Corollary 4.** *Let $\mathbf{Z}_1^T$ be a sequence of random variables. Then, for any $\delta > 0$, with probability at least $1 - \delta$, the following holds for all $f \in \mathcal{F}$ and all $\alpha > 0$:*

$$\mathbb{E}[f(Z_{T+1})|\mathbf{Z}_1^T] \leq$$
$$\sum_{t=1}^{T} q_t f(Z_t) + \widetilde{\Delta} + \Delta_s + 4\alpha + M \big[ \|\mathbf{q}\|_2 + \|\mathbf{q} - \mathbf{p}\|_2 \big] \sqrt{2 \log \frac{2 \, \mathbb{E}_{\mathbf{v} \sim T(\mathbf{p})}[\mathcal{N}_1(\alpha, \mathcal{G}, \mathbf{z})]}{\delta}},$$

*where $\widetilde{\Delta} = \sup_{f \in \mathcal{F}} \left( \sum_{t=1}^{T} (p_t - q_t) f(Z_t) \right)$.*

## 5  Algorithms

In this section, we use our learning guarantees to devise algorithms for forecasting non-stationary time series. We consider a broad family of kernel-based hypothesis classes with regression losses. We present the full analysis of this setting in Appendix B including novel bounds on the sequential Rademacher complexity. The learning bounds of Theorem 1 can be generalized to hold uniformly over $\mathbf{q}$ at the price of an additional term in $O\Big( \|\mathbf{q} - \mathbf{u}\|_1 \sqrt{\log_2 \log_2 \|\mathbf{q} - \mathbf{u}\|_1^{-1}} \Big)$. We prove this result in Theorem 8 (Appendix B). Suppose $L$ is the squared loss and $H = \{ \mathbf{x} \to \mathbf{w} \cdot \Psi(\mathbf{x}) \colon \|\mathbf{w}\|_{\mathcal{H}} \leq \Lambda \}$, where $\Psi \colon \mathcal{X} \to \mathcal{H}$ is a feature mapping from $\mathcal{X}$ to a Hilbert space $\mathcal{H}$. By Lemma 6 (Appendix B), we can bound the complexity term in our generalization bounds by

$$O\Big( (\log^3 T) \frac{\Lambda r}{\sqrt{T}} + (\log^3 T) \|\mathbf{q} - \mathbf{u}\|_1 \Big),$$

where $K$ is a PDS kernel associated with $\mathcal{H}$ such that $\sup_x K(x, x) \leq r$ and $\mathbf{u}$ is the uniform distribution over the sample. Then, we can formulate a joint optimization problem over both $\mathbf{q}$ and $\mathbf{w}$ based on the learning guarantee of Theorem 8, which holds uniformly over all $\mathbf{q}$:

$$\min_{0 \leq \mathbf{q} \leq 1, \mathbf{w}} \left\{ \sum_{t=1}^{T} q_t (\mathbf{w} \cdot \Psi(x_t) - y_t)^2 + \lambda_1 \sum_{t=1}^{T} d_t q_t + \lambda_2 \|\mathbf{w}\|_{\mathcal{H}}^2 + \lambda_3 \|\mathbf{q} - \mathbf{u}\|_1 \right\}. \tag{4}$$

Here, we have upper bounded the empirical discrepancy term by $\sum_{t=1}^{T} d_t q_t$ with each $d_t$ defined by $\sup_{\mathbf{w}' \leq \Lambda} | \sum_{s=1}^{T} p_s (\mathbf{w}' \cdot \Psi(x_s) - y_s)^2 - (\mathbf{w}' \cdot \Psi(x_t) - y_t)^2 |$. Each $d_t$ can be precomputed using DC-programming. For general loss functions, the DC-programming approach only guarantees convergence to a stationary point. However, for the squared loss, our problem can be cast as an instance of the trust region problem, which can be solved globally using the DCA algorithm of Tao and An [1998]. Note that problem (4) is not jointly convex in $\mathbf{q}$ and $\mathbf{w}$. However, using the dual problem associated to $\mathbf{w}$ yields the following equivalent problem, it can be rewritten as follows:

$$
\min_{0 \leq \mathbf{q} \leq 1} \left\{ \max_{\boldsymbol{\alpha}} \left\{ -\lambda_2 \sum_{t=1}^{T} \frac{\alpha_t^2}{q_t} - \boldsymbol{\alpha}^T \mathbf{K} \boldsymbol{\alpha} + 2\lambda_2 \boldsymbol{\alpha}^T \mathbf{Y} \right\} + \lambda_1 (\mathbf{d} \cdot \mathbf{q}) + \lambda_3 \|\mathbf{q} - \mathbf{u}\|_1 \right\}, \qquad (5)
$$

where $\mathbf{d} = (d_1, \ldots, d_T)^T$, $\mathbf{K}$ is the kernel matrix and $\mathbf{Y} = (y_1, \ldots, y_T)^T$. We use the change of variables $r_t = 1/q_t$ and further upper bound $\lambda_3 \|\mathbf{q} - \mathbf{u}\|_1$ by $\lambda_3' \|\mathbf{r} - T^2 \mathbf{u}\|_2$, which follows from $|q_t - u_t| = |q_t u_t (r_t - T)|$ and Hölder's inequality. Then, this yields the following optimization problem:

$$
\min_{\mathbf{r} \in \mathcal{D}} \left\{ \max_{\boldsymbol{\alpha}} \left\{ -\lambda_2 \sum_{t=1}^{T} r_t \alpha_t^2 - \boldsymbol{\alpha}^T \mathbf{K} \boldsymbol{\alpha} + 2\lambda_2 \boldsymbol{\alpha}^T \mathbf{Y} \right\} + \lambda_1 \sum_{t=1}^{T} \frac{d_t}{r_t} + \lambda_3 \|\mathbf{r} - T^2 \mathbf{u}\|_2^2 \right\}, \qquad (6)
$$

where $\mathcal{D} = \{\mathbf{r} \colon r_t \geq 1, t \in [1, T]\}$. The optimization problem (6) is convex since $\mathcal{D}$ is a convex set, the first term in (6) is convex as a maximum of convex (linear) functions of $\mathbf{r}$. This problem can be solved using standard descent methods, where, at each iteration, we solve a standard QP in $\boldsymbol{\alpha}$, which admits a closed-form solution. Parameters $\lambda_1$, $\lambda_2$, and $\lambda_3$ are selected through cross-validation.

An alternative simpler algorithm based on the data-dependent bounds of Corollary 4 consists of first finding a distribution $\mathbf{q}$ minimizing the (regularized) discrepancy and then using that to find a hypothesis minimizing the (regularized) weighted empirical risk. This leads to the following two-stage procedure. First, we find a solution $\mathbf{q}^*$ of the following convex optimization problem:

$$
\min_{\mathbf{q} \geq 0} \left\{ \sup_{\mathbf{w}' \leq \Lambda} \left( \sum_{t=1}^{T} (p_t - q_t)(\mathbf{w}' \cdot \Psi(x_t) - y_t)^2 \right) + \lambda_1 \|\mathbf{q} - \mathbf{u}\|_1 \right\}, \qquad (7)
$$

where $\lambda_1$ and $\Lambda$ are parameters that can be selected via cross-validation. Our generalization bounds hold for arbitrary weights $\mathbf{q}$ but we restrict them to being positive sequences. Note that other regularization terms such as $\|\mathbf{q}\|_2^2$ and $\|\mathbf{q} - \mathbf{p}\|_2^2$ from the bound of Corollary 4 can be incorporated in the optimization problem, but we discard them to minimize the number of parameters. This problem can be solved using standard descent optimization methods, where, at each step, we use DC-programming to evaluate the supremum over $\mathbf{w}'$. Alternatively, one can upper bound the supremum by $\sum_{t=1}^{T} q_t d_t$ and then solve the resulting optimization problem.

The solution $\mathbf{q}^*$ of (7) is then used to solve the following (weighted) kernel ridge regression problem:

$$
\min_{\mathbf{w}} \left\{ \sum_{t=1}^{T} q_t^* (\mathbf{w} \cdot \Psi(x_t) - y_t)^2 + \lambda_2 \|\mathbf{w}\|_{\mathcal{H}}^2 \right\}. \qquad (8)
$$

Note that, in order to guarantee the convexity of this problem, we require $\mathbf{q}^* \geq 0$.

## 6 Conclusion

We presented a general theoretical analysis of learning in the broad scenario of non-stationary non-mixing processes, the realistic setting for a variety of applications. We discussed in detail several algorithms benefitting from the learning guarantees presented. Our theory can also provide a finer analysis of several existing algorithms and help devise alternative principled learning algorithms.

**Acknowledgments**

This work was partly funded by NSF IIS-1117591 and CCF-1535987, and the NSERC PGS D3.

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
