[Supplementary Material]

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

$ with joint distribution $\mathbf{p}$, let $\mathbf{Z}'^T_1$ be a decoupled tangent sequence. Then, for any measurable function $G$, the following equality holds*

$$\mathbb{E}\left[G\Big(\sup_{f\in\mathcal{F}}\sum_{t=1}^{T}q_t(f(Z'_t)-f(Z_t))\Big)\right] = \mathbb{E}_{\boldsymbol{\sigma}}\mathbb{E}_{\mathbf{z}\sim T(\mathbf{p})}\left[G\Big(\sup_{f}\sum_{t=1}^{T}\sigma_t q_t(f(\mathbf{z}'_t(\boldsymbol{\sigma}))-f(\mathbf{z}_t(\boldsymbol{\sigma})))\Big)\right]. \quad (9)$$

*The result also holds with the absolute value around the sums in (9).*

*Proof.* The proof follows an argument in the proof of Theorem 3 of [Rakhlin et al., 2011]. We only need to check that every step holds for an arbitrary weight vector $\mathbf{q}$, in lieu of the uniform distribution vector $\mathbf{u}$, and for an arbitrary measurable function $G$, instead of the identity function. Observe that we can write the left-hand side of (9) as

$$\mathbb{E}\left[G\Big(\sup_{f\in\mathcal{F}}\Sigma(\boldsymbol{\sigma})\Big)\right] = \mathbb{E}_{Z_1,Z'_1\sim\mathbf{p}_1}\mathbb{E}_{Z_2,Z'_2\sim\mathbf{p}_2(\cdot|Z_1)}\cdots\mathbb{E}_{Z_T,Z'_T\sim\mathbf{p}_T(\cdot|\mathbf{Z}_1^{T-1})}\left[G\Big(\sup_{f\in\mathcal{F}}\Sigma(\boldsymbol{\sigma})\Big)\right],$$

where $\boldsymbol{\sigma} = (1,\ldots,1) \in \{\pm1\}^T$ and $\Sigma(\boldsymbol{\sigma}) = \sum_{t=1}^{T}\sigma_t q_t(f(Z'_t) - f(Z_t))$. Now, by definition of decoupled tangent sequences, the value of the last expression is unchanged if we swap the sign of any $\sigma_{i-1}$ to $-1$ since that is equivalent to permuting $Z_i$ and $Z'_i$. Thus, the last expression is in fact equal to

$$\mathbb{E}_{Z_1,Z'_1\sim\mathbf{p}_1}\mathbb{E}_{Z_2,Z'_2\sim\mathbf{p}_2(\cdot|S_1(\sigma_1))}\cdots\mathbb{E}_{Z_T,Z'_T\sim\mathbf{p}_T(\cdot|S_1(\sigma_1),\ldots,S_{T-1}(\sigma_{T-1}))}\left[G\Big(\sup_{f\in\mathcal{F}}\Sigma(\boldsymbol{\sigma})\Big)\right]$$

for any sequence $\boldsymbol{\sigma} \in \{\pm1\}^T$, where $S_t(1) = Z_t$ and $Z'_t$ otherwise. Since this equality holds for any $\boldsymbol{\sigma}$, it also holds for the mean with respect to uniformly distributed $\boldsymbol{\sigma}$. Therefore, the last expression is equal to

$$\mathbb{E}_{\boldsymbol{\sigma}}\mathbb{E}_{Z_1,Z'_1\sim\mathbf{p}_1}\mathbb{E}_{Z_2,Z'_2\sim\mathbf{p}_2(\cdot|S_1(\sigma_1))}\cdots\mathbb{E}_{Z_T,Z'_T\sim\mathbf{p}_T(\cdot|S_1(\sigma_1),\ldots,S_{T-1}(\sigma_{T-1}))}\left[G\Big(\sup_{f\in\mathcal{F}}\Sigma(\boldsymbol{\sigma})\Big)\right].$$

This last expectation coincides with the expectation with respect to drawing a random tree $\mathbf{z}$ from $T(\mathbf{p})$ (and its tangent tree $\mathbf{z}'$) and a random path $\boldsymbol{\sigma}$ to follow in that tree. That is, the last expectation is equal to

$$\mathbb{E}_{\boldsymbol{\sigma}}\mathbb{E}_{\mathbf{z}\sim T(\mathbf{p})}\left[G\Big(\sup_{f}\sum_{t=1}^{T}\sigma_t q_t(f(\mathbf{z}'_t(\boldsymbol{\sigma})) - f(\mathbf{z}_t(\boldsymbol{\sigma})))\Big)\right],$$

which concludes the proof. $\qquad\square$

**Theorem 3.** *Let $\mathbf{Z}_1^T$ be a sequence of random variables. Then, for any $\delta > 0$, with probability at least $1 - \delta$, the following holds for all $\alpha > 0$:*

$$\sup_{f\in\mathcal{F}}\left(\sum_{t=1}^{T}(p_t - q_t)\,\mathbb{E}[f(Z_t)|\mathbf{Z}_1^{t-1}]\right) \leq \sup_{f\in\mathcal{F}}\left(\sum_{t=1}^{T}(p_t - q_t)f(Z_t)\right)$$
$$+ \alpha + M\|\mathbf{q} - \mathbf{p}\|_2\sqrt{\log\frac{\mathbb{E}_{\mathbf{z}\sim T(\mathbf{p})}[\mathcal{N}_1(\alpha,\mathcal{G},\mathbf{z})]}{\delta'}},$$

*where $\mathbf{p}$ is the distribution the uniform on the last $s$ points.*

*Proof.* First, observe that

$$\sup_{f\in\mathcal{F}}\left(\sum_{t=1}^{T}(p_t - q_t)\,\mathbb{E}[f(Z_t)|\mathbf{Z}_1^{t-1}]\right) - \sup_{f\in\mathcal{F}}\left(\sum_{t=1}^{T}(p_t - q_t)f(Z_t)\right)$$
$$\leq \sup_{f\in\mathcal{F}}\left(\sum_{t=1}^{T}(p_t - q_t)(\mathbb{E}[f(Z_t)|\mathbf{Z}_1^{t-1}] - f(Z_t))\right).$$

The result then follows using similar arguments to those used in the proof of Theorem 1. $\qquad\square$

# B  Generalization Bounds for Kernel-Based Hypotheses with Regression Losses

In this section, we present generalization bounds for kernel-based hypothesis with regression losses. One of the main technical tools used in our analysis is the notion of *sequential Rademacher complexity*. Let $\mathcal{G}$ be a set of functions from $\mathcal{Z}$ to $\mathbb{R}$. The sequential Rademacher complexity of a function class $\mathcal{Z}$ is defined as the following:

$$\mathfrak{R}_T^{\text{seq}}(\mathcal{G}) = \sup_{\mathbf{z}} \mathbb{E}\left[\sup_{g \in \mathcal{G}} \sum_{t=1}^{T} \sigma_t q_t g(z_t(\boldsymbol{\sigma}))\right], \tag{10}$$

where the supremum is taken over all complete binary trees of depth $T$ with values in $\mathcal{Z}$ and where $\boldsymbol{\sigma}$ is a sequence of Rademacher random variables. This a combinatorial measure of complexity which makes bounds based on this notion coarser than those of Theorem 1, which are stated in terms of expected covering numbers. However, it turns out that this coarser analysis is sufficient for the derivation of our algorithms in Section 5. We also remark that most of the results in this section can be tightened using the notion of *distribution-dependent* Rademacher complexity, but we defer these results to the full version of the paper.

Our first result is a bound on the sequential Rademacher complexity of the kernel-based hypothesis with regression losses.

**Lemma 6.** *Let $p \geq 1$ and $\mathcal{F} = \{(\mathbf{x}, y) \to (\mathbf{w} \cdot \Psi(\mathbf{x}) - y)^p \colon \|\mathbf{w}\|_{\mathcal{H}} \leq \Lambda\}$ where $\mathcal{H}$ is a Hilbert space and $\Psi \colon \mathcal{X} \to \mathcal{H}$ a feature map. Assume that the condition $|\mathbf{w} \cdot \mathbf{x} - y| \leq M$ holds for all $(\mathbf{x}, y) \in \mathcal{Z}$ and all $\mathbf{w}$ such that $\|\mathbf{w}\|_{\mathcal{H}} \leq \Lambda$. Then, the following inequalities hold:*

$$\mathfrak{R}_T^{seq}(\mathcal{F}) \leq pM^{p-1}C_T\mathfrak{R}_T^{seq}(H) \leq C_T\left(pM^{p-1}\frac{\Lambda r}{\sqrt{T}} + pM^p\|\mathbf{q} - \mathbf{u}\|_1\right), \tag{11}$$

*where $K$ is a PDS kernel associated to $\mathcal{H}$, $H = \{\mathbf{x} \to \mathbf{w} \cdot \Psi(\mathbf{x}) : \|\mathbf{w}\|_{\mathcal{H}} \leq \Lambda\}$, $r = \sup_x K(x, x)$, and $C_T = 8(1 + 4\sqrt{2}\log^{3/2}(eT^2))$.*

*Proof.* We begin the proof by setting $q_t f(\mathbf{z}_t(\boldsymbol{\sigma})) = q_t(\mathbf{w} \cdot \Psi(\mathbf{x}_t(\sigma)) - \mathbf{y}_t(\sigma))^2 = \frac{1}{T}(\mathbf{w} \cdot \mathbf{x}'(\boldsymbol{\sigma}) - \mathbf{y}'_t)^2$, where $\mathbf{x}'_t(\boldsymbol{\sigma}) = \sqrt{Tq_t}\Psi(\mathbf{x}_t(\sigma))$ and $\mathbf{y}'_t(\boldsymbol{\sigma}) = \sqrt{Tq_t}\mathbf{y}_t(\boldsymbol{\sigma})$. We let $\mathbf{z}'_t = (\mathbf{x}'_t, \mathbf{y}'_t)$. Then we observe that

$$\mathfrak{R}_T^{\text{seq}}(\mathcal{F}) = \sup_{\mathbf{z}'=(\mathbf{x}',\mathbf{y}')} \mathbb{E}_{\boldsymbol{\sigma}}\left[\sup_{\mathbf{w}} \frac{1}{T}\sum_{t=1}^{T} \sigma_t(\mathbf{w} \cdot \mathbf{x}'_t(\boldsymbol{\sigma}) - \mathbf{y}'_t(\boldsymbol{\sigma}))^p\right]$$

$$= \sup_{\mathbf{z}=(\mathbf{x},\mathbf{y})} \mathbb{E}_{\boldsymbol{\sigma}}\left[\sup_{\mathbf{w}} \sum_{t=1}^{T} q_t\sigma_t(\mathbf{w} \cdot \mathbf{x}_t(\boldsymbol{\sigma}) - \mathbf{y}_t(\boldsymbol{\sigma}))^p\right].$$

Since $x \to |x|^p$ is $pM^{p-1}$-Lipschitz over $[-M, M]$, by Lemma 13 in [Rakhlin et al., 2015], the following bound holds:

$$\mathfrak{R}_T^{\text{seq}}(\mathcal{F}) \leq pM^{p-1}C_T\mathfrak{R}_T^{\text{seq}}(H'),$$

where $H' = \{(\mathbf{x}, y) \to \mathbf{w} \cdot \Psi(\mathbf{x}) - y : \|\mathbf{w}\|_{\mathcal{H}} \leq \Lambda\}$. Note that Lemma 13 requires that $\mathfrak{R}_T^{\text{seq}}(H') > 1/T$ which is guaranteed by Khintchine's inequality. By definition of the sequential Rademacher complexity

$$\mathfrak{R}_T^{\text{seq}}(H') = \sup_{(\mathbf{x},y)} \mathbb{E}_{\boldsymbol{\sigma}}\left[\sup_{\mathbf{w}} \sum_{t=1}^{T} \sigma_t q_t(\mathbf{w} \cdot \Psi(\mathbf{x}_t(\boldsymbol{\sigma})) - y(\boldsymbol{\sigma}))\right]$$

$$= \sup_{\mathbf{x}} \mathbb{E}_{\boldsymbol{\sigma}}\left[\sup_{\mathbf{w}} \sum_{t=1}^{T} \sigma_t q_t\mathbf{w} \cdot \Psi(\mathbf{x}_t(\boldsymbol{\sigma}))\right] + \sup_{y} \mathbb{E}_{\boldsymbol{\sigma}}\left[\sum_{t=1}^{T} \sigma_t q_t y(\boldsymbol{\sigma})\right] = \mathfrak{R}_T^{\text{seq}}(H),$$

where for the last equality we used the fact that $\sigma_t$s are mean zero random variables and $\sigma_t$ is independent of $y(\boldsymbol{\sigma}) = y(\sigma_1, \sigma_2, \ldots, \sigma_{t-1})$. This proves the first result. To prove the second bound we first observe that

$$\mathfrak{R}_T^{\text{seq}}(H) \leq \frac{1}{T}\sup_{\mathbf{x}} \mathbb{E}_{\boldsymbol{\sigma}}\left[\sup_{\mathbf{w}} \sum_{t=1}^{T} \sigma_t\mathbf{w} \cdot \Psi(\mathbf{x}_t(\boldsymbol{\sigma}))\right] + M\|\mathbf{q} - \mathbf{u}\|_1.$$

Next, the first term on the right-hand side can be bounded as follows:

$$\frac{1}{T}\sup_{\mathbf{x}}\mathbb{E}_{\boldsymbol{\sigma}}\left[\sup_{\mathbf{w}}\sum_{t=1}^{T}\sigma_t\mathbf{w}\cdot\Psi(\mathbf{x}_t(\boldsymbol{\sigma}))\right] \leq \frac{\Lambda}{T}\sup_{\mathbf{x}}\mathbb{E}_{\boldsymbol{\sigma}}\left\|\sum_{t=1}^{T}\sigma_t\Psi(\mathbf{x}_t(\boldsymbol{\sigma}))\right\|_{\mathcal{H}}$$

$$\leq \frac{\Lambda}{T}\sup_{\mathbf{x}}\sqrt{\mathbb{E}_{\boldsymbol{\sigma}}\left\|\sum_{t=1}^{T}\sigma_t\Psi(\mathbf{x}_t(\boldsymbol{\sigma}))\right\|_{\mathcal{H}}^2}$$

$$= \frac{\Lambda}{T}\sup_{\mathbf{x}}\sqrt{\mathbb{E}_{\boldsymbol{\sigma}}\left[\sum_{t,s=1}^{T}\sigma_t\sigma_s\Psi(\mathbf{x}_t(\boldsymbol{\sigma}))\cdot\Psi(\mathbf{x}_s(\boldsymbol{\sigma}))\right]}$$

$$\leq \frac{\Lambda}{T}\sup_{\mathbf{x}}\sqrt{\sum_{t=1}^{T}\mathbb{E}_{\boldsymbol{\sigma}}[K(x_t(\boldsymbol{\sigma}),x_t(\boldsymbol{\sigma}))]}$$

$$\leq \frac{\Lambda r}{\sqrt{T}},$$

where again we are using the fact that if $s < t$ then

$$\mathbb{E}_{\boldsymbol{\sigma}}[\sigma_t\sigma_s K(x_t(\sigma),x_s(\sigma))] = \mathbb{E}_{\boldsymbol{\sigma}}[\sigma_t]\,\mathbb{E}_{\boldsymbol{\sigma}}[\sigma_s K(x_t(\sigma),x_s(\sigma))] = 0$$

by the independence of $\sigma_t$ from $\sigma_s$, $x_t(\sigma) = x_t(\sigma_1,\ldots,\sigma_{t-1})$ and $x_s(\sigma) = x_s(\sigma_1,\ldots,\sigma_s)$. $\qquad\square$

Our next result establishes a high-probability learning guarantee for kernel-based hypothesis.

**Theorem 7.** *Let $p \geq 1$ and $\mathcal{F} = \{(\mathbf{x},y) \to (\mathbf{w}\cdot\Psi(\mathbf{x}) - y)^p\colon \|\mathbf{w}\|_{\mathcal{H}} \leq \Lambda\}$ where $\mathcal{H}$ is a Hilbert space and $\Psi\colon \mathcal{X} \to \mathcal{H}$ a feature map. Assume that the condition $|\mathbf{w}\cdot\mathbf{x} - y| \leq M$ holds for all $(\mathbf{x},y) \in \mathcal{Z}$ and all $\mathbf{w}$ such that $\|\mathbf{w}\|_{\mathcal{H}} \leq \Lambda$. If $\mathbf{Z}_1^T = (\mathbf{X}_1^T, \mathbf{Y}_1^T)$ is a sequence of random variables then, for any $\delta > 0$, with probability at least $1 - \delta$ the following holds for all $h \in \{\mathbf{x} \to \mathbf{w}\cdot\Psi(\mathbf{x})\colon \|\mathbf{w}\|_{\mathcal{H}} \leq \Lambda\}$:*

$$\mathbb{E}[(h(X_{T+1}) - Y_{T+1})^p|\mathbf{Z}_1^T] \leq \sum_{t=1}^{T}(h(X_t) - Y_t)^p + \Delta$$

$$+ M\widetilde{C}_T\sqrt{\log\frac{8L}{\delta}}\left(pM^{p-1}\frac{\Lambda r}{\sqrt{T}} + pM^p\|\mathbf{q} - \mathbf{u}\|_1\right),$$

*where $L = \max\left\{e^4, \sum_{j=1}^{\infty}\mathcal{N}_\infty(2^{-j},\mathcal{F})\right\}$, $\widetilde{C}_T = c\log^{3/2}T(1 + 4\sqrt{2}\log^{3/2}(eT^2))$ and where $c$ is an absolute constant. Thus, for $p = 2$,*

$$\mathbb{E}[(h(X_{T+1}) - Y_{T+1})^2|\mathbf{Z}_1^T] \leq \sum_{t=1}^{T}(h(X_t) - Y_t)^2 + \Delta + O\left((\log^3 T)\frac{\Lambda r}{\sqrt{T}} + (\log^3 T)\|\mathbf{q} - \mathbf{u}\|_1\right).$$

Note that for this result to be non-trivial we need $\sum_{j=1}^{\infty}\mathcal{N}_\infty(2^{-j},\mathcal{F}) < \infty$. This condition is easy to verify in our case. First, observe that for any set of linear functions the inequality $\mathcal{N}_\infty(\alpha, H) > \Lambda r/\alpha$ holds and it follows that $\sum_{j=1}^{\infty}\mathcal{N}_\infty(2^{-j},\mathcal{F}) < 2\Lambda r$. The case of composition of $H$ with $\ell_p$ loss can be handled by realizing that this composition leads to a linear function in a higher dimensional space corresponding to a polynomial kernel of degree $p$.

*Proof.* The beginning of the proof closely follows that of Theorem 1. The first step is to observe that since the difference of the suprema is bounded by the supremum of the difference, it suffices to bound the probability of the following event

$$\left\{\sup_{f\in\mathcal{F}}\left(\sum_{t=1}^{T}q_t(\mathbb{E}[f(Z_t)|\mathbf{Z}_1^{t-1}] - f(Z_t))\right) \geq \epsilon\right\}.$$

Next, we note that $q_t f(Z_t) = q_t(\mathbf{w} \cdot \Psi(X_t) - Y_t)^2 = \frac{1}{T}(\mathbf{w} \cdot X_t' - Y_t')^2$, where $X_t' = \sqrt{Tq_t}\Psi(X_t)$ and $Y_t' = \sqrt{Tq_t}Y_t$. We let $Z_t' = (X_t', Y_t')$. Applying Lemma 15 from Rakhlin et al. [2015], we obtain that for any $\delta > 0$ with probability at least $1 - \delta$, the following holds for all $f \in \mathcal{F}$:

$$\Phi(\mathbf{Z}_1^T) - \Delta \leq M \mathfrak{R}_T^{\text{seq}}(\mathcal{F})(\log^{3/2} T)\sqrt{c\log\frac{8L}{\delta}},$$

where

$$\mathfrak{R}_T^{\text{seq}}(\mathcal{F}) = \sup_{\mathbf{z}'=(\mathbf{x}',\mathbf{y}')} \mathbb{E}_{\boldsymbol{\sigma}}\left[\sup_{\mathbf{w}}\frac{1}{T}\sum_{t=1}^{T}\sigma_t(\mathbf{w}\cdot\mathbf{x}_t'(\boldsymbol{\sigma}) - \mathbf{y}_t'(\boldsymbol{\sigma}))^p\right]$$

$$= \sup_{\mathbf{z}=(\mathbf{x},\mathbf{y})} \mathbb{E}_{\boldsymbol{\sigma}}\left[\sup_{\mathbf{w}}\sum_{t=1}^{T}q_t\sigma_t(\mathbf{w}\cdot\mathbf{x}_t(\boldsymbol{\sigma}) - \mathbf{y}_t(\boldsymbol{\sigma}))^p\right]$$

is the sequential Rademacher complexity of $\mathcal{F}$. Note that $\mathfrak{R}_T^{\text{seq}}(\mathcal{F}) > 1/T$ as in the proof of Lemma 6. The desired result follows from Lemma 6. $\qquad\square$

The final result of this section extends Theorem 7 to hold uniformly over $\mathbf{q}$s.

**Theorem 8.** *Let $p \geq 1$ and $\mathcal{F} = \{(\mathbf{x}, y) \to (\mathbf{w} \cdot \Psi(\mathbf{x}) - y)^p \colon \|\mathbf{w}\|_{\mathcal{H}} \leq \Lambda\}$ where $\mathcal{H}$ is a Hilbert space and $\Psi\colon \mathcal{X} \to \mathcal{H}$ a feature map. Assume that the condition $|\mathbf{w}\cdot\mathbf{x} - y| \leq M$ holds for all $(\mathbf{x}, y) \in \mathcal{Z}$ and all $\mathbf{w}$ such that $\|\mathbf{w}\|_{\mathcal{H}} \leq \Lambda$. Then, if $\mathbf{Z}_1^T = (\mathbf{X}_1^T, \mathbf{Y}_1^T)$ is a sequence of random variables, for any $\delta > 0$, with probability at least $1 - \delta$, the following bound holds for all $h \in H = \{\mathbf{x} \to \mathbf{w}\cdot\Psi(\mathbf{x})\colon \|\mathbf{w}\|_{\mathcal{H}} \leq \Lambda\}$ and all $\mathbf{q}$ such that $0 < \|\mathbf{q} - \mathbf{u}\|_1 \leq 1$:*

$$\mathbb{E}[(h(X_{T+1}) - Y_{T+1})^p | \mathbf{Z}_1^T] \leq \sum_{t=1}^{T}(h(X_t) - Y_t)^p + \Delta + 4M\|\mathbf{q} - \mathbf{u}\|_1$$

$$+ M\widetilde{C}_T\Big(\sqrt{\log\frac{16L}{\delta}} + \sqrt{\log\log_2 2\|\mathbf{q} - \mathbf{u}\|^{-1}}\Big)\Big(pM^{p-1}\frac{\Lambda r}{\sqrt{T}} + pM^p\|\mathbf{q} - \mathbf{u}\|_1\Big),$$

*where $\widetilde{C}_T = c\log^{3/2} T(1 + 4\sqrt{2}\log^{3/2}(eT^2))$, $c$ is an absolute constant and $L = \max\{e^4, \sum_{j=1}^{\infty}\mathcal{N}_{\infty}(2^{-j}, \mathcal{F})\}$. Thus, for $p = 2$,*

$$\mathbb{E}[(h(X_{T+1}) - Y_{T+1})^2 | \mathbf{Z}_1^T] \leq \sum_{t=1}^{T}(h(X_t) - Y_t)^2 + \Delta$$

$$+ O\Big((\log^3 T)\sqrt{\log\log_2 2\|\mathbf{q} - \mathbf{u}\|^{-1}}\Big(\frac{\Lambda r}{\sqrt{T}} + \|\mathbf{q} - \mathbf{u}\|_1\Big)\Big).$$

This result suggests that we should try to minimize $\sum_{t=1}^{T} q_t f(Z_t) + \Delta$ over $\mathbf{q}$ and $\mathbf{w}$ making sure that $\mathbf{q}$ does not deviate from $\mathbf{u}$ by more than $O(T^{-1/2})$. Theorem 1 can be extended in a similar way to hold uniformly over $\mathbf{q}$s and we will provide this result in the full version of the paper.

*Proof.* Let $(\epsilon_k)_{k=0}^{\infty}$ and $(\mathbf{q}(k))_{k=0}^{\infty}$ be infinite sequences specified below. By Theorem 7, the following holds for each $k$

$$\mathbb{P}\bigg(\mathbb{E}[f(Z_{T+1})|\mathbf{Z}_1^T] > \sum_{t=1}^{T}q_t(k)f(Z_t) + \Delta(\mathbf{q}(k)) + C\epsilon_k\bigg) \leq 8L\exp(-\epsilon_k^2),$$

where $\Delta(\mathbf{q}(k))$ denotes the discrepancy computed with respect to the weights $\mathbf{q}(k)$ and $C$ is equal to

$$M\widetilde{C}_T\Big(pM^{p-1}\frac{\Lambda r}{\sqrt{T}} + pM^p\|\mathbf{q}(k) - \mathbf{u}\|_1\Big).$$

Table 1: Average squared error (standard deviation)

|  | ads1 | ads2 | ads3 |
|---|---|---|---|
| DBF | **0.0001 (0.0001)** | **0.0002 (0.0001)** | **0.0047 (0.0001)** |
| WAR | 0.0099 (0.0155) | 0.0997 (0.1449) | 0.1026 (0.1509) |
| ARIMA | 0.1432 (0.2091) | 0.4797 (0.6942) | 0.2598 (0.3696) |

Let $\epsilon_k = \epsilon + \sqrt{2 \log k}$. Then, by the union bound we can write

$$\mathbb{P}\left(\exists k\colon \mathbb{E}[f(Z_{T+1})|\mathbf{Z}_1^T] > \sum_{t=1}^{T} q_t(k)f(Z_t) + \Delta(\mathbf{q}(k)) + C\epsilon_k\right) \leq \sum_{k=1}^{\infty} 8L \exp(-\epsilon_k^2)$$

$$\leq \sum_{k=1}^{\infty} 8L \exp(-\epsilon^2 - \log k^2)$$

$$\leq 16L \exp(-\epsilon^2).$$

We choose the sequence $\mathbf{q}(k)$ to satisfy $\|\mathbf{q}(k) - \mathbf{u}\|_1 = 2^{-k}$. Then, for any $\mathbf{q}$ such that $0 < \|\mathbf{q} - \mathbf{u}\|_1 \leq 1$, there exists $k$ such that

$$\|\mathbf{q}(k) - \mathbf{u}\|_1 < \|\mathbf{q} - \mathbf{u}\|_1 \leq \|\mathbf{q}(k-1) - \mathbf{u}\|_1 = 2\|\mathbf{q}(k) - \mathbf{u}\|_1.$$

Thus, the following inequality holds:

$$\sqrt{2 \log k} \leq \sqrt{2 \log \log_2 2\|\mathbf{q} - \mathbf{u}\|_1^{-1}}.$$

Combining this with the observation that the following two inequalities hold:

$$\sum_{t=1}^{T} q_t(k)f(Z_t) \leq \sum_{t=1}^{T} q_t f(Z_t) + 2M\|\mathbf{q} - \mathbf{u}\|_1$$

$$\Delta(\mathbf{q}(k)) \leq \Delta(\mathbf{q}) + 2M\|\mathbf{q} - \mathbf{u}\|_1,$$

shows that the event

$$\left\{\mathbb{E}[f(Z_{T+1})|\mathbf{Z}_1^T] > \sum_{t=1}^{T} q_t f(Z_t) + \Delta + C\left(\epsilon + \sqrt{2 \log \log_2 2\|\mathbf{q} - \mathbf{u}\|_1^{-1}}\right) + 4M\|\mathbf{q} - \mathbf{u}\|_1\right\}$$

implies the following one

$$\left\{\mathbb{E}[f(Z_{T+1})|\mathbf{Z}_1^T] > \sum_{t=1}^{T} q_t(k)f(Z_t) + \Delta(\mathbf{q}(k)) + C\epsilon_k\right\},$$

which completes the proof. $\qquad\square$

## C  Experiments

In Section 5, we described an algorithm benefitting from our learning guarantees based on solving the convex optimization problem (6). Due to submission time constraints, our experiments were carried out instead by solving directly problem (4) using an alternating optimization method. This is based on the observation that for a fixed $\mathbf{q}$, problem (4) is a simple QP over $\mathbf{w}$ and, for a fixed $\mathbf{w}$, the problem reduces to an LP in $\mathbf{q}$. This suggests an iterative scheme where we alternate between each of these two problems.

We have compared our algorithm against a standard ARIMA model that is commonly used in practice for forecasting non-stationary time series, as well as a weighted autoregression algorithm (WAR) that solves optimization problem in (8) with $\mathbf{q}$ tuned manually.

In our experiments, we have used three artificial datasets: ads1, ads2, ads3. For each dataset, we have generated time series with 2,000 sample points, trained on the first 1,999 points and tested on

the last point. To gain statistically significance, we repeat this procedure $1,000$ times. To generate these time series the following autoregressive processes have been used:

$$\texttt{ads1:} \quad Y_t = \alpha_t Y_{t-1} + \epsilon_t, \quad \alpha_t = 1 \text{ if } t < 1800 \text{ and } -1 \text{ otherwise},$$
$$\texttt{ads2:} \quad Y_t = \alpha_t Y_{t-1} + \epsilon_t, \quad \alpha_t = 0.9 - 1.8(t/2000),$$
$$\texttt{ads3:} \quad Y_t = \alpha_t Y_{t-1} + (1-\alpha_t)Y_{t-2} + \epsilon_t, \quad \alpha_t = 0.9t/2000,$$

where $\epsilon_t$ are independent standard Gaussian random variables.

The results of our experiments are summarized in Table 1. Observe that the results of each experiment are statistically significant using paired $t$-test and in each case our discrepancy-based forecaster (DBF) significantly outperforms other algorithms. Moreover, DBF has a better performance on at least 90% of individual runs in each experiment.

## D  Optimization Problem

In this section, we provide a detailed derivation of the optimization problem in (6) starting with optimization problem in (4). The first step is to appeal to the following chain of equalities:

$$\min_{\mathbf{w}} \left\{ \sum_{t=1}^{T} q_t (\mathbf{w} \cdot \Psi(x_t) - y_t)^2 + \lambda_2 \|\mathbf{w}\|_{\mathcal{H}}^2 \right\}$$
$$= \min_{\mathbf{w}} \left\{ \sum_{t=1}^{T} (\mathbf{w} \cdot x_t' - y_t')^2 + \lambda_2 \|\mathbf{w}\|_{\mathcal{H}}^2 \right\}$$
$$= \max_{\boldsymbol{\beta}} \left\{ -\lambda_2 \sum_{t=1}^{T} \beta_t^2 - \sum_{s,t=1}^{T} \beta_s \beta_t x_s' x_t' + 2\lambda_2 \sum_{t=1}^{T} \beta_t y_t' \right\}$$
$$= \max_{\boldsymbol{\beta}} \left\{ -\lambda_2 \sum_{t=1}^{T} \beta_t^2 - \sum_{s,t=1}^{T} \beta_s \beta_t \sqrt{q_s}\sqrt{q_t} K_{s,t} + 2\lambda_2 \sum_{t=1}^{T} \beta_t \sqrt{q_t} y_t \right\}$$
$$= \max_{\boldsymbol{\alpha}} \left\{ -\lambda_2 \sum_{t=1}^{T} \frac{\alpha_t^2}{q_t} - \boldsymbol{\alpha}^T \mathbf{K} \boldsymbol{\alpha} + 2\lambda_2 \boldsymbol{\alpha}^T \mathbf{Y} \right\}, \tag{12}$$

where the first equality follows by substituting $x_t' = \sqrt{q_t}\Psi(x_t)$ and $y_t' = \sqrt{q_t}y_t$ the second equality uses the dual formulation of the kernel ridge regression problem and the last equality follows from the following change of variables: $\alpha_t = \sqrt{q_t}\beta_t$.

By (12), optimization problem in (4) is equivalent to the following optimization problem

$$\min_{0 \leq \mathbf{q} \leq 1} \left\{ \max_{\boldsymbol{\alpha}} \left\{ -\lambda_2 \sum_{t=1}^{T} \frac{\alpha_t^2}{q_t} - \boldsymbol{\alpha}^T \mathbf{K} \boldsymbol{\alpha} + 2\lambda_2 \boldsymbol{\alpha}^T \mathbf{Y} \right\} + \lambda_1 (\mathbf{d} \cdot \mathbf{q}) + \lambda_3 \|\mathbf{q} - \mathbf{u}\|_1 \right\}.$$

Next, we apply the change of variables $r_t = 1/q_t$, and upper bound the last term in the objective $\lambda_3 \|\mathbf{q} - \mathbf{u}\|_1$ by $\lambda_3' \|\mathbf{r} - T^2 \mathbf{u}\|_2$, where we use the fact that $|q_t - u_t| = |q_t u_t (r_t - T)|$ and Hölder's inequality. This leads to the following convex optimization problem:

$$\min_{\mathbf{r} \in \mathcal{D}} \left\{ \max_{\boldsymbol{\alpha}} \left\{ -\lambda_2 \sum_{t=1}^{T} r_t \alpha_t^2 - \boldsymbol{\alpha}^T \mathbf{K} \boldsymbol{\alpha} + 2\lambda_2 \boldsymbol{\alpha}^T \mathbf{Y} \right\} + \lambda_1 \sum_{t=1}^{T} \frac{d_t}{r_t} + \lambda_3'' \|\mathbf{r} - T^2 \mathbf{u}\|_2 \right\}.$$

This optimization problem is convex, since the domain $\mathcal{D} = \{\mathbf{r} \colon r_t \geq 1, \forall t \in [1, T]\}$ and the first term in the objective is a maximum of convex (linear) functions of $\mathbf{r}$ and hence is a convex function of $\mathbf{r}$. The last term in the objective is equivalent to a constraint $\|\mathbf{r} - T^2 \mathbf{u}\|_2 \leq \Lambda$ or $\|\mathbf{r} - T^2 \mathbf{u}\|_2^2 \leq \Lambda^2$, for some $\Lambda$. This allows us to write the optimization equivalently as

$$\min_{\mathbf{r} \in \mathcal{D}} \left\{ \max_{\boldsymbol{\alpha}} \left\{ -\lambda_2 \sum_{t=1}^{T} r_t \alpha_t^2 - \boldsymbol{\alpha}^T \mathbf{K} \boldsymbol{\alpha} + 2\lambda_2 \boldsymbol{\alpha}^T \mathbf{Y} \right\} + \lambda_1 \sum_{t=1}^{T} \frac{d_t}{r_t} + \lambda_3'' \|\mathbf{r} - T^2 \mathbf{u}\|_2^2 \right\},$$

which is exactly the problem in (6).