[Reviews · NeurIPS 2015]

Submitted by Assigned_Reviewer_1

Summary:

This paper studies learning with time series with quite relaxed assumptions on the stochastic process defining the time series: No mixing or stationarity assumptions is required.

This is achieved by the use of two concepts:

1) Sequential covering number, which roughly speaking is an extension of the standard covering number to a norm defined over binary tree structures, and 2) Discrepancy, which is a measure of the non-stationarity of the process.

The paper provides a high probability deviation inequality that is a function of the covering number and discrepancy. It suggests a method for estimating the discrepancy, under some extra assumptions. It also introduces some algorithms for forecasting using the obtained upper bound.

Even though having an upper bound that is valid for non-stationary and non-mixing processes might seem almost magical, one might say that the trick is that one can move all the apparent complexity of the process into the discrepancy and the sequential covering number. If the process is too difficult to learn, these quantities would be too large.

The interesting point here, however, is that the quantities that measure the complexity of learning are defined at the right level. To understand my comment, notice that the usual approach to study learning for time series is that we start by analyzing the difficulty of stochastic process (through concepts such as mixing) and then under certain assumptions on the stochastic process, we convert it to the usual i.i.d. setting using techniques such as independent blocks. One can then define covering number/Rademacher complexity/etc. over these independent blocks, and use the standard tools that work for independent samples. This approach separates the study of complexity of the stochastic process and the complexity of function/hypothesis space, and may lead to pessimistic results.

What this paper does is that it defines the complexity at the joint stochastic process-function space level. This can potentially lead to much tighter results.

The key elements of this approach are not necessarily new.

Sequential complexities have been defined by Rakhlin, Sridharan, Tewari (NIPS 2010, NIPS 2011, JMLR 2015). A similar discrepancy measure has also been defined and used by Kuznetsov and Mohri (ALT 2014), Mohri and Munoz Medina (ALT 2012), and even before that in the domain adaptation literature.

The use of these together, however, is novel as far as I know.

Evaluation:

The topic of the paper is interesting and important, the approach is elegant and as far I can tell novel, and the paper itself is more or less well-written. I have some comments below and I encourage the authors to answer and/or consider them.

* L142: Is q_t missing in the LHS?

* Is optimizing only the discrepancy term of the upper bound of Corollary 3 the right thing to do?

Corollary 2 shows that the upper bound depends on the choice of q. In the beginning of Section 4, it is suggested that one should minimize the discrepancy as a function of q and using it to weigh the training samples.

If I understand correctly, the covering number also depends on q (as the cover is defined w.r.t. q-weighted l_1 norm). If that is the case, shouldn't it be considered in the optimization too? (I understand that this might be computationally very difficult or impossible.)

* Please compare your algorithms with those in Kuznetsov and Mohri, "Forecasting Non-Stationary Time Series: From Theory to Algorithms," NIPS 2014 Workshop on Transfer and Multi-task Learning. Montreal, Canada, December 2014.

* Is it possible to have Bernstein-like bound in Theorem 1?

* L222: It seems that the first inequality can actually be an equality.

* L246: It might be more clear to use c_t(z_t(\sigma)) instead of c_t(\sigma).

* L251: exp(lambda alpha) -> exp( 2 lambda alpha)

* L256: c_T(c) -> c_t(sigma)

* Some examples of the behaviour of E[N_1(alpha,G,v)] would be helpful. For example, if the process has a certain mixing property, is it different from when it is i.i.d.?

* L137: "definition definition" -> "definition"

* L177: "leg" -> "lag"

* L339-342: The discussion is a bit vague here.

* L350: delta' -> delta

* L492 (Appendix A): Maybe it is more clear to say that (z,z') is from T(p).

* L598: Isn't the change from q to the uniform u too loose? This leads to M || q - u ||_1, which kind of ignores all the structure in the function space except its boundedness.
Summary: The topic of the paper is interesting and important, the approach is elegant and as far I can tell novel, and the paper itself is more or less well-written

Submitted by Assigned_Reviewer_2

This paper studies the general non-stationary non-mixing processes and provides the first generalization bounds in this setting. The generalization bounds depends on the proposed measure of discrepancy, which can be estimated from data with some mild assumptions. Base on the guarantees, the authors show algorithms

for forecasting non-stationary time series.

The problem of forecasting non-stationary non-mixing stochastic processes is important and interesting, and it is a nice extension to prior works that mostly assume stationarity or mixing. The proposed data-dependent bounds are novel and neat.

The results and proofs seem correct, but maybe due to space limit, it is sometimes too dense and may not be easily readable by readers not familiar with this topic. For example, the term "DC-programming" is first used in line 385 without even mentioning its full-name.

Some other questions and comments: 1) In line 125,

the authors states that q_1,.., a_T are real numbers. Do they need to sum up to one? If not, then in line 170, \Delta is not necessarily 0.

2) It would be good if the authors can add some intuitive explanation of equation (1), and why it is important, since it is the key quantity of interest in the analysis.

Typos: equation (1), missed a ")" line 413, can selected = > can be selected
Summary: This paper provides data-dependent learning bounds for the non-stationary non-mixing stochastic processes. The theoretical results are novel and the paper is well-written.

Submitted by Assigned_Reviewer_3

This paper provides a new generalization bound for quantifying time series prediction error using the notion of the supremum of an empirical process. At the core of the result is a term of discrepancy measure for the time series. It quantifies the accuracy of prediction, and is a very intuitive and natural mathematical characterization.

In summary, I found the paper very interesting and containing original results. However, for saying that, I have to mention that the paper clearly sets a relatively high bar for readers to understand, with a lot of concepts treated as known to readers and stated without carefully introduced.

Some technical comments are as follows: (i) The authors need to discuss the boundedness assumption on the loss function, which seems too strong to me.

(ii) The authors highlight the generalization of this paper to nonstationary and nonmixing time series. So a natural question is: how sharp the result is? For example, if the time series is, say, actually strongly mixing, can we recover the corresponding results?

(iii) Regarding the main theorem, I failed to find any constraint on the function class "G" and real value "alpha". Intuitively, should G be the function class "F" in line 122?

Some minor comments: (iv) l74: give (v) l137: definition definition (vi) l186: [K and M, 2014]

Summary: The paper provides new bounds for quantifying prediction error in time series analysis. The result is interesting in my personal opinion.

Submitted by Assigned_Reviewer_4

The paper provides generalisation bounds for prediction of time series with side information. The bounds are very technical and the discussion is rather terse (which can be due to space limitations).

The bounds are used to propose a kernel algorithm, which is compares against standard time-series methods in the appendix.

Minor issues:

Page 3, displayed eq (1): a closing bracket is missing.

Page 3, line -22: the word "definition" is repeated.

Page 6, Corollary 2. I suspect there may be a confusion of T vs T+1.
Summary: The paper provides generalisation bounds for prediction of time series with side information.

Author Feedback
Author rebuttal: We thank the reviewers for carefully reading our paper and for their helpful comments. We provide answers to the main questions below and we will address the rest in the final version.

Reviewer_1:

- the reviewer is asking weather the complexity measure should also be considered in optimization problem discussed at the beginning of Section 4. The answer is yes and the algorithms of Section 5 include this complexity via regularization terms. We will improve the discussion in Section 4 to clarify this point.

- the reviewer would like us to compare our algorithms with the work of (Kuznetsov & Mohri, 2014): in short, their algorithms are based either on a non-convex optimization problem or require setting q_ts manually. In contrast, our formulation is convex and allows finding optimal q_ts automatically.

- we thank the reviewer for numerous other insightful comments. We will include the necessary corrections and clarifications in the final version.

Reviewer_3:

- the reviewer is asking about the boundedness assumption on the loss function. This is a standard assumption in learning theory. We do believe that it can be relaxed however using the notion of offset Rademacher complexity (Rakhlin & Sridharan, 2014).

- the reviewer is asking how sharp our bounds are with respect to different assumptions. We provide a brief discussion addressing this question at the end of Section 3. We will expand on that in the final version.

- the reviewer is asking about G and alpha in Theorem 1. G should be replaced by F and we have the following condition: 0 < 2*alpha < epsilon. We will fix that in the final version.

Reviewer_5:

- the reviewer is asking about restrictions on q_ts. Indeed, on line 125 it is assumed that q_ts form a probability distribution. However, many other results in this paper do not require this assumption.

Reviewer_7: we are grateful to the reviewer for his appreciation of this work.

Reviewer_8:

- the reviewer is concerned that finding discrepancies d_ts, required as inputs for our algorithm, involves solving a non-convex DC program. For general loss functions, the DC-programming approach only guarantees convergence to a critical point. However, for squared loss, our problem can be cast as an instance of the trust region problem that can be solved globally using DC algorithm (Tao & An, 1998).

We emphasize that once d_ts are determined, optimization problem (6) that is solved by our algorithm is convex.

Reviewer_9: we thank the reviewer for his interest in our work.